# KLF4, Slug and EMT in Head and Neck Squamous Cell Carcinoma

**DOI:** 10.3390/cells10030539

**Published:** 2021-03-03

**Authors:** Julia Ingruber, Dragana Savic, Teresa Bernadette Steinbichler, Susanne Sprung, Felix Fleischer, Rudolf Glueckert, Gabriele Schweigl, Ira-Ida Skvortsova, Herbert Riechelmann, József Dudás

**Affiliations:** 1Department of Otorhinolaryngology and Head and Neck Surgery, Medical University of Innsbruck, University Hospital of Tyrol, A-6020 Innsbruck, Austria; julia.ingruber@i-med.ac.at (J.I.); teresa.steinbichler@i-med.ac.at (T.B.S.); felix.fleischer@i-med.ac.at (F.F.); rudolf.glueckert@i-med.ac.at (R.G.); gabi.schweigl@tirol-kliniken.at (G.S.); herbert.riechelmann@i-med.ac.at (H.R.); 2Laboratory for Experimental and Translational Research on Radiation Oncology (EXTRO-Lab), Department of Therapeutic Radiology and Oncology, Medical University of Innsbruck, A-6020 Innsbruck, Austria; dragana.savic@i-med.ac.at (D.S.); ira.skvortsova@i-med.ac.at (I.-I.S.); 3Tyrolean Cancer Research Institute, A-6020 Innsbruck, Austria; 4Department of Pathology, Medical University of Innsbruck, A-6020 Innsbruck, Austria; susanne.sprung@i-med.ac.at; 5Department of Restorative and Operative Dentistry, Medical University of Innsbruck, A-6020 Innsbruck, Austria

**Keywords:** TissueFaxs, TWIST, ZEB, SCC-25, FaDu, UPCI-SCC090

## Abstract

Epithelial to mesenchymal transition (EMT) is clinically relevant in head and neck squamous cell carcinoma (HNSCC). We hypothesized that EMT-transcription factors (EMT-TFs) and an anti-EMT factor, Krüppel-like-factor-4 (KLF4) regulate EMT in HNSCC. Ten control mucosa and 37 HNSCC tissue samples and three HNSCC cell lines were included for investigation of EMT-TFs, KLF4 and vimentin at mRNA and protein levels. Slug gene expression was significantly higher, whereas, KLF4 gene expression was significantly lower in HNSCC than in normal mucosa. In the majority of HNSCC samples, there was a significant negative correlation between KLF4 and Slug gene expression. Slug gene expression was significantly higher in human papilloma virus (HPV) negative HNSCC, and in tumor samples with irregular p53 gene sequence. Transforming-growth-factor-beta-1 (TGF- β1) contributed to downregulation of KLF4 and upregulation of Slug. Two possible regulatory pathways could be suggested: (1) EMT-factors induced pathway, where TGF-β1 induced Slug together with vimentin, and KLF4 was down regulated at the same time; (2) p53 mutations contributed to upregulation and stabilization of Slug, where also KLF4 could co-exist with EMT-TFs.

## 1. Introduction

A reversible program of trans-differentiations from the epithelial cell type to a mesenchymal endpoint (epithelial–to-mesenchymal-transition; EMT) is crucial for embryonic development. The first descriptions of EMT originate from Elisabeth Hay and her colleagues [1], who was also the first who discussed the role of transforming growth factor-beta (TGF-β) in induction of EMT [2].

Interestingly, EMT is reactivated in cancer, but a full transition from an epithelial cell type to a fibroblastic mesenchymal one is rare [3,4]. The incomplete EMT process in cancer cells is accompanied by the activation of EMT-transcription factors (EMT-TFs): TWIST, ZEB1-2, SLUG [3], which participate in increased cancer cell motility, allow dissemination of individual tumor cells or collective migration of cell clusters [3], but also have pleiotropic roles, that are not exclusively required for EMT [5]. EMT is associated with a core transcription signature, which is frequently reported as the down-regulation of E-cadherin and the up-regulation of SNAIL, SLUG, ZEB, and TWIST EMT-TFs. The detection of EMT-TFs in cancer tissue is negatively correlated with survival [3,6]. EMT-TFs, as it was reported for Slug, function as master regulators and allow cells to enter into the tumor-initiating cell state [7,8]. Moreover, experimental overexpression of Slug promoted self-renewal of HNSCC cells and increased the gene expression of stem cell markers [8]. 

In addition to the EMT-TFs (SNAIL, SLUG, ZEB, and TWIST) discussed in major original and review papers, other transcription factors also regulate epithelial or mesenchymal gene expression profiles and contribute to the epithelial or mesenchymal phenotype, as for example the group of zinc finger transcription factors: the Krüppel-like factors (KLFs). KLF4, for example, is associated with epithelial differentiation and is down-regulated by TGF-β1 during EMT in hepatocellular carcinoma [9]. In normal tissue, KLF4 is detectable in the nuclei of terminally differentiated epithelial cells [10,11], whereas, its expression is frequently lost in various human cancer types, such as colorectal cancer, gastric cancer, esophageal squamous cell carcinoma [12]. In tumor tissue not only the KLF4 levels are lower compared to normal tissue, but its expression is inversely correlated with the stage of tumors and survival [12,13]. Tiwari et al. published in 2013 that down-regulation of KLF4 expression is required for the induction of EMT in vitro, and for metastasis in vivo [13]. 

In head and neck squamous cell carcinoma (HNSCC) tumor microenvironment (TME) components as cancer-associated fibroblasts (CAFs), immune and other supporting cells significantly influence the outcome of the disease, for example by production of cytokines as TGF-β1. TGF-β1 is the primary factor triggering EMT in HNSCC [14]. In addition, Slug was published to have major regulatory roles in EMT of HNSCC [15]. Not only Slug, but the above-mentioned KLF4 was discussed in HNSCC before.

Part of the HNSCC cases had maintained or increased KLF4 gene expression, another part had decreased KLF4 gene expression compared to normal reference tissue. The persistent KLF4 gene expression correlated with a worse disease-specific survival [16]. In HNSCC two important factors, human papilloma virus infection and the mutational status of the tumor suppressor gene TP53 seem to significantly influence patient survival and prognosis.

In contrast to the classically discussed risk factors as tobacco use or alcohol consumption, also human papilloma virus (HPV), a sexually transmitted infection, contributes significantly to this disease [17]. Moreover, HPV-positive tumors are not only different in terms of their etiology, but they respond better both to radio(chemo)therapy [18], and to surgery [19], compared to HPV-negative tumors. Consequently, HPV-positive patients have better prognosis. The association of the wild type p53 to HPV-driven HNSCC was reported to contribute to this favorable prognosis [20]. Non-silent p53 mutations are not frequent in HPV-positive HNSCC [21], whereas they are present in 35% of HPV-negative tumors [21]. The non-silent p53 mutations in HNSCC are associated with worse prognosis. Previous reports also discussed the relationship between the regulation of EMT and the mutational status of p53 [22].

The previously presented background encouraged us to set up a hypothesis, which sees the EMT-transcription factors, especially the members of the SNAI family as EMT driver factors on one side, and an anti-EMT factor, KLF4, an epithelial-differentiation-related transcriptional regulator, on the other side. 

Our recent published work [23] evidenced that Slug immunohistochemistry using a mouse monoclonal antibody (clone S43-1259) is a utilizable alternative to complicated EMT determination methods, and Slug was suggested as a surrogate EMT-marker in head and neck cancer (HNC) [23]. Nevertheless, Slug, as the main EMT-TF in HNSCC, its significance and its presence still raise several questions, which will be addressed in this work. One important point was published by Ye et al. from the research group of Robert A. Weinberg [7]. They found that Slug was expressed in normal basal mammary epithelial cells, whereas Snail, as well as Twist and Zeb1 were present in stromal fibroblasts. Furthermore, authors argue that Snail, but not Slug is associated with induction of mesenchymal phenotype in breast cancer cells [7]. At RNA level, using specific primers and real-time PCR it was possible to distinguish between Snail and Slug. Unlike in breast cancer, in HNSCC Snail is very low expressed, and it is not significantly induced in experimental conditions. In contrast, Slug is present in both normal mucosa and in HNSCC, and the main difference between normal and tumor tissue in terms of Slug expression is quantitative. In further steps, we also investigated the gene expression of TWIST and ZEB in HNSCC. 

In this study we hypothesize, that (1) Slug expression as main EMT-TF is upregulated in EMT cells of HNSCC, whereas, (2) KLF4, as anti-EMT TF is lost in the head and neck cancer cells undergoing EMT. (3) We further suggest, that for these changes the main EMT-inducer, TGF-β is responsible. 

## 2. Materials and Methods

### 2.1. Patient Samples for RNA Isolation

Oropharyngeal control mucosa was obtained from 10 patients during uvulopalatopharyngoplasty (UPPP), a routine surgical procedure for the treatment of snoring and obstructive sleep apnea [24]. The intact mucosa was confirmed histologically by pathologists. The tissue samples of 37 HNSCC patients (patient data listed in Table 1) were surgically removed during pan-endoscopy following the patients ‘consent and in agreement with our ethic approval (UN4428, ethic commission meeting of 303/4,14, 26 July 2011), and used for RNA isolation, reverse transcription and real-time RT-PCR.

### 2.2. RNA Isolation, Reverse Transcription, PCR and Sequencing

For RNA isolation 2–3 mm tissue slices were collected and lysed in 1 mL TRIzol^®^ Reagent (Ambion^®^, Life Technologies^TM^, Carlsbad, CA, USA), and RNA was isolated as instructed by the manufacturer. RNA concentrations were determined by fluorometric measurements (Qubit, Invitrogen, Darmstadt, Germany), and RNA quality and integrity were identified by Qubit RNA IQ kit (Invitrogen). The proportion of intact RNA of total RNA isolates was at least 70%. Two micrograms of total RNA were reverse transcribed by M-MuLV Reverse Transcriptase with 2 micrograms of oligo dT_15_ (GeneON, Ludwigshafen am Rhein, Germany) in a ThermoQ heating and cooling block (Biozym, Hessisch Oldendorf, Germany). 

Specific sequences of primers used are detailed in Appendix A. The primers were synthesized by Invitrogen, Darmstadt, Germany and were used for real-time PCR utilizing the Sensifast Sybr Fluorescein Kit of Bioline (Labconsulting, Vienna, Austria) and the Bio-Rad MyiQ^TM^ (Bio-Rad, Laboratories, Inc., Hercules, CA, USA) cycler according to the manufacturer’s protocol. 

GAPDH was used as housekeeping gene, and relative quantities of SNAI1, SLUG, ZEB1, TWIST and KLF4 transcripts were calculated by pair-wise differences of threshold cycles (∆_CT_) of gene of interest and the loading control housekeeping gene [25]. According to Livak et al. [25], in the final analysis, we used the relative quantification and related the PCR signal in both HNSCC and control mucosa to a reference, which was the mean value of the control uvula samples from UPPP. The identity of the PCR products of genes discussed in this study were confirmed by Sanger sequencing by Microsynth Austria (Vienna, Austria). 

### 2.3. Immunohistochemistry and Immunofluorescence Staining, Image Acquisition

Indirect immunofluorescence staining was performed by Ventana Discovery Classic Immunostainer, using pre-diluted mouse monoclonal pan-cytokeratin antibody and an anti-vimentin, clone VI-10 mouse monoclonal antibody, and a rabbit monoclonal anti-KLF4 (Table 2). Secondary anti-mouse IgM or IgG and anti-rabbit IgG Alexa Fluor^TM^ 488 or 594, or TRITC conjugated antibodies for detection of the immunoreaction were purchased from Molecular Probes (Life Technologies, Darmstadt, Germany) and were used at 1:200 final dilution. All antibodies were diluted as suggested by the manufacturers. Negative control stainings were done using isotype matching mouse IgG1, IgM and rabbit IgG antibodies. The immunofluorescent-stained (IFS) slides were cell nuclei counterstained by DAPI (Molecular Probes), autofluorescence was removed by the TrueView^TM^ autofluorescence quenching kit and slides were covered in Vectashield Vibrance (both from Vector Laboratories, Burlingame, CA, USA). Immunofluorescence reactions were acquired in a TissueFaxs system (TissueGnostics, Vienna, Austria), using a PCO Pixelfly CCD monochrome camera (PCO Inc., Kelheim, Germany). 

Enzyme immunohistochemistry for Slug was done using the mouse monoclonal anti-Slug antibody, and the reaction was developed by universal secondary antibody (Ventana) and the DAB Map kit of Ventana. Further staining experiments were performed with anti-KLF4 antibody and anti-TGFβ1 antibody (Table 2). 

The immunohistochemistry labelling levels were acquired in TissueFaxs system in brightfield using a firewire connected Pixelink camera (Pixelink, Rochester, NY, USA). The acquired images were imported in HistoQuest program and the staining signal was quantified using a single-reference-shade colour deconvolution algorithm [23]. The staining signal was a mixture of brown reaction and blue counterstain of haematoxylin, which could also present itself as clear blue colour or clear brown colour as complete negative or full positive reactions. The analysis was based in the HistoQuest program on counting of positive cells, which were recognized on the basis of “blue cell nuclei”. All tissue sections were acquired after setting the homogenous illumination by Köhler, and white balance was also set before saving the lamp intensity and exposure time. Moreover, all tissue sections were acquired using exactly the same acquisition profile. Based on the properties of the original tissue, different threshold level intensities were identified in technical control tissue sections stained with mouse monoclonal isotype control primary antibody, and therefore, cut off values 7–11 were used to define Slug positive cells. 

The passage “Appendix A”, contains further details on distinguishing between Snail and Slug antibody reactions. 

### 2.4. Cell Lines, Cell Culture and Treatment Conditions 

SCC-25 and UPCI-SCC090 cells were acquired from the German Collection of Microorganisms and Cell Cultures (DSMZ, Braunschweig, Germany, DSMZ no.: ACC 617). FaDu cells were purchased from the American Type Culture Collection (ATCC), Manassas, VA, USA.

SCC-25 cells were cultured in DMEM/Ham’s F12 medium, UPCI-SCC090 [26] cells in EMEM medium supplemented with 10% FBS, 2 mM l-glutamine, 100 units/mL penicillin and 100 μg/mL streptomycin [27,28,29,30,31]. FaDu cells were cultured in Minimum Essential Medium (Eagle) with Earle’s BSS (Merck Millipore, Vienna, Austria). Culture medium was supplemented with 2 mM L-glutamine, 1.5 g/L sodium bicarbonate, 0.1 mM non-essential amino acids ((Gibco, Life Technologies, Grand Island, NY, USA), 100 U/mL penicillin, 100 µg/mL streptomycin, and 10% fetal calf serum (Merck Millipore, Vienna, Austria). For experimental purposes all cells were cultured in an albumin-containing corresponding culture medium where serum proteins were replaced by 4.4 g/L bovine serum albumin from PAA Laboratories, (Pasching, Austria).

For the treatment with 50 ng/mL IL-6 [32] or with 1 ng/mL TGF-β1 [33] (RnD Systems, Minneapolis, MN, USA), 5 × 10^4^ cells/mL were plated on 12-well cell culture plates (BD Falcon, Vienna, Austria) in 1 mL albumin-containing medium per well [33] supplemented with the treatment factors and cultured for 72 h, followed by replacement of the medium and treatments for an additional 48 h. After the end of the treatment procedure cells of 4 wells were scraped together into 500 µL of RIPA buffer (50 mM Tris HCl/pH:7.4, 1 mM EDTA, 0.5 mM EGTA, 1% Triton X-100, 0.25% sodium deoxycholate, 0.1% sodium dodecylsulfate, 150 mM NaCl, 10 mM NaF, 1 mM PMSF; 10 µL proteinase inhibitors of Invitrogen “Halt Inhibitors mix”/mL RIPA buffer)/culture dish. The cell suspension was vortexed and incubated 3-times for 15 min on ice, homogenized in 22G needles and centrifuged at 15,000× *g*, 15 min, 4 °C. 

One experimental set contained two-times three plates, one control, one TGF-β1-treated and one IL-6-treated plate. All plates contained 3 samples. The experiment was repeated in two sets, and finally contained 12 control, 12 TGF-β1-treated and 12 IL-6-treated samples. 

### 2.5. Western Blot

The cleared supernatant was subjected to protein concentration measurement using the Pierce 660 nm protein assay (Pierce, Rochford, IL, USA) according to the instructions of the manufacturer. 10 µg protein from all samples was subsequently processed for western blot, using Invitrogen NuPage gels, electrophoresis and blotting system. Western blot detection was done as published before [27], using primary antibodies: rabbit monoclonal anti-Slug, mouse monoclonal anti-Slug at 125 ng/mL, rabbit monoclonal anti-vimentin, mouse monoclonal anti-GAPDH, rabbit monoclonal anti-KLF4, rabbit monoclonal anti-p38 MAPK and rabbit monoclonal anti-phospho-p38 MAPK, rabbit polyclonal anti-Smad2/3, rabbit monoclonal anti-phospho-Smad2, rabbit monoclonal anti-Stat3 and rabbit monoclonal anti-phospho-Stat3 (Table 3). For signal detection horseradish peroxidase coupled matched secondary antibodies (1:1000 and 1:2000 dilutions, depending on the instruction of the manufacturer) and chemiluminescent substrate of Azure Biosystems (Biomedica, Vienna, Austria) were used in conditions suggested by the manufacturer. The chemilumescence signal was imaged by an Azure C500 documentation system (Biomedica, Vienna, Austria). GAPDH was detected using 1:10,000 diluted near infrared fluorescence conjugates labelled and anti-mouse IgG (cat. nr. 926-68180) secondary antibody, available from Li-Cor Biosciences, Bad Homburg, Germany. 

The optical density (OD) of western blot bands was measured by using the Li-cor Image Studio Lite software. The ODs of vimentin, Slug and KLF4 proteins were normalized by the ODs of GAPDH. The normalized ODs were related to the ODs of the control samples in the experimental set. The distribution of the relative normalized ODs of Slug, KLF4 and vimentin was statistically tested by the D’Agostino and Pearson omnibus normality test. The differences in the three groups (control, TGF-β1- or IL-6-treated) were compared by parametric or non-parametric ANOVA, followed by pairwise comparisons with the control data set by Mann-Whitney test or unpaired Student’s *t*-test depending on the normal or non-parametric distribution of the data sets. 

## 3. Results

### 3.1. Gene Expression Analysis of EMT-Related Transcription Factors in HNSCC Compared with Normal Oral Mucosa

Ten control normal mucosa of UPPP and 37 HNSCC tumor tissue samples were included in the real-time PCR analysis. Following our hypothesis, the EMT-transcription factors: SNAI1, SLUG, ZEB1 and TWIST, as well as the “anti-EMT” transcription factor KLF4 were investigated. The CT-values of GAPDH housekeeping gene in all samples were by 20–22, and did not show sample-specific regulation (Appendix A). 

The CT-values of Slug ranged from 29 to 31 in control normal mucosa, and from 21 to 31 in HNSCC. All samples provided valid specific PCR products. Typical PCR products of the investigated genes were sent for Sanger sequencing and the specificity of the PCR reaction was confirmed. In this regard, Slug was clearly expressed in both control mucosa and HNSCC, and the CT values suggested a higher expression in tumor tissue. No significant difference of gene expression in HNSCC related to normal mucosa was detected for ZEB1. TWIST PCR did not produce evaluable specific products.

The CT-values of KLF4 ranged from 27 to 31 in control normal mucosa, and from 22 to 33 in HNSCC. All samples provided valid specific PCR products (Appendix A). 

If control mucosa and HNSCC were compared, the median of Slug gene expression was significantly higher in HNSCC (*p* = 0.009), whereas, the median of KLF4 was significantly lower in HNSCC than in normal mucosa (*p* = 0.041) (Figure 1). These results fulfilled the expectations based on previous publications [12]. Nevertheless, as visible on Figure 1, some HNSCC cases had lower Slug and higher KLF4 gene expression than the normal mucosa reference level. 

Tai and colleagues published in 2011 that there are two groups of HNSCC tissue samples. In the first larger group (70% of HNSCC samples), the KLF4 gene expression decreases compared to the surrounding normal epithelium. In the second smaller group, the KLF4 gene expression remains persistent and comparable to the surrounding normal epithelium [16]. In our set of HNSCC samples we compared the KLF4 gene expression with that of the normal epithelium from normal mucosa obtained by UPPP. In 29 of 37 HNSCC samples available for this analysis, KLF4 gene expression decreased compared to that of normal mucosa (Figure 2). In 8 of 37 HNSCC samples KLF4 gene expression remained persistent. In 3 samples both KLF4 and Slug were upregulated (not shown). In the samples where KLF4 was decreased, Slug gene expression was upregulated and there was a significant negative correlation between KLF4 and Slug gene expression (Spearman r: −0.3625; *p* = 0.0253) (Figure 2). In the samples where KLF4 remained persistent, Slug was not upregulated, and there was no significant negative correlation between KLF4 and Slug gene expression (not shown).

In a further step, Slug and KLF4 gene expression in HNSCC with and without human papilloma virus background and with regular and irregular p53 gene background were investigated.

### 3.2. Gene Expression of Slug and KLF4 in HNSCC in Relation to HPV and p53 Background

HPV-positivity was determined immunohistochemically by being in at least 66% of the tumor cells p16^INK4^positive [34]. Taking HPV DNA PCR analysis as the reference method, the sensitivity of p16 IHC is 78% and the specificity is 79% [35]. As previously published by our clinic, the HPV-positive cases show significantly better survival (*p* = 0.015 by Log-Rank (Mantel-Cox) pairwise comparison) [36].

A scattered TP53 staining (using the diagnostic antibody clone Bp53-11, [36]) is related with normal (wild type) genetic background with no p53 mutations [37], while no staining or increased (over 66% of tumor cells stained) staining pattern is related with “altered”, frequently even mutated p53 [37]. We amplified the complete protein coding region of p53 mRNA and sequenced it. Thereby we found a statistically significant correlation between confirmed p53 sequence mutations or mRNA loss and irregular staining pattern. Irregular gene expression consists of sequencing confirmed mutations and lack of gene product, which is also confirmed by PCR. A scattered, regular p53 staining pattern and wild type p53 mRNA sequence were also related (not shown, Spearman R: 0.617; *p* < 10^−4^).

In HPV^−^ HNSCC Slug gene expression was significantly higher than in HPV^+^ (Figure 3a). KLF4 gene expression at mRNA level was not statistically different in HPV^+^ and HPV^−^ HNSCC (Figure 3b). In HNSCC with irregular p53 immunostaining pattern and sequence changes (mutations) in the p53 coding region the Slug gene expression was significantly higher than in HNSCC with regular p53 (Figure 3c). KLF4 did not show a significant gene expression difference in relation to p53 genetical background (Figure 3d). 

### 3.3. Distribution of Slug KLF4 and TGF-β1 in HNSCC Tumor Tissue

In addition to gene expression, the protein Slug, and KLF4 were detected immunohistochemically in HNSCC. KLF4 was detectable both in fluorescence and in enzyme immunohistochemistry, but Slug was detectable only in enzyme immunohistochemistry (Figure 4). Although, a significant correlation between immunohistochemical Slug staining intensity or the frequency of the stained cells among tumor cells and Slug gene expression was not possible to state, both the % of Slug positive cells in cancer cell nests as well as the Slug staining intensity were higher in HNSCC cases with upregulated Slug gene expression than in the cases with Slug gene expression level at the normal control mucosa (Appendix A). 

In control normal mucosa (Figure 4a,b) the layered squamous epithelial cells contained cytokeratin (CK) in their cytoplasm and KLF4 in their cell nuclei (Figure 4a). No traces of Slug were detected in this tissue by enzyme immunohistochemistry (Figure 4b). In HPV^+^ oropharynx carcinoma with wild type p53 mRNA coding region sequence the cytoplasmatic CK was scattered in the cancer cell nests, whereas, KLF4 was represented in all tumor cell nuclei (Figure 4c). As in the normal epithelium, also in this tissue, there were no traces of Slug immunolabeling (Figure 4d). In HPV^−^ HNSCC with irregular p53 coding sequences Slug immunostaining was detected (Figure 4f,h), the CK staining was heterogeneous in the cytoplasm of the tumor cells (Figure 4e,g), KLF4 could be intensive positive or almost completely lost in the cell nuclei of the tumor cells (Figure 4e,g), and vimentin traces could be also present in the cancer cell nests (Figure 4e). The tissue in Figure 4g,h is presented in more details in Figure 5.

In this more detailed image clustered Slug^+^ cells are displayed in cancer cell nests in the tumor-stroma interface (Figure 5a). In this interface scattered stroma cells showed strong TGF-β1 reaction, as well as few tumor cells were labelled with TGF-β1 antibody (Figure 5b). This interface area was also investigated using immunofluorescence by increased magnification, and several cells with combined reaction of KLF4, cytokeratin and vimentin were detected, which might be considered as EMT tumor cells (Figure 5c). Similarly to the HNSCC sample presented in Figure 5 (with p53 mutation), also the HNSCC sample with lost p53 gene product showed intensive Slug-staining, TGF-β1 reaction confined to scattered stroma cells at the tumor—stroma interface, as well as tumor cells which displayed vimentin combined with cytokeratin and/or with KLF4 (Figure 6). 

In HNSCC with irregular p53 genetic background and immunohistochemistry, Slug was present either in cell cluster or diffusely in the cancer cell nests, and several tumor cells of Slug^+^ cancer cell nests were combined positive for CK, vimentin and KLF4. TGF-β1 was hypothesized as an inducer of Slug or vimentin or EMT in HNSCC. Looking at the TGF-β1 immunohistochemical reaction presented in Figure 5b and Figure 6b, it could be concluded, that although, TGF-β1 was present at the tumor/stroma interface, its levels were low. Interestingly, as presented in Appendix A, peritumoral vimentin^+^ stroma-fibroblasts (Appendix A) might produce high amounts of TGF-β1 (Appendix A), this is still not enough for induction of Slug, or for presence of cells containing epithelial and mesenchymal marker at the same time. 

### 3.4. Higher Slug, and Vimentin, and Decreased KLF4 Protein Levels in HNSCC Cell Line Models Treated with TGF-β1

The involvement of TGF-β1 and IL-6 as EMT-inducers, in the regulation of the protein products of KLF4, Slug and VIM (Vimentin) were investigated in control and 1 ng/mL TGF-β1 or 50 ng/mL IL-6—treated conditions in HNSCC cell lines. The western blot analysis showed significant regulations in SCC-25 cells and UPCI-SCC-90 cells. Slug, KLF4 and vimentin were clearly detectable in SCC-25 cells using high sensitive chemiluminescence (Figure 7a). TGF-β1 induced a significant increase (*p* = 10^−4^ by Kruskal-Wallis test and Dunn’s multiple comparison) of Slug protein levels (Figure 7a,c) and significant reduction (*p* = 0.017 by unpaired *t*-test compared to control) of KLF4 protein levels (Figure 7b,d). Similarly, IL-6 upregulated Slug at protein level, but it was not statistically significant. IL-6 did not induce a significant downregulation of KLF4 in SCC25-cells.

Vimentin was detected at protein level using chemiluminescence in control conditions of SCC-25 cells and, both TGF-β1 and IL-6 induced a significant increase (*p* = 0.0083 Kruskal-Wallis test and Dunn’s multiple comparison) (Figure 7e,f).

In FaDu cells, KLF4 was constitutive present and was not significantly regulated by TGF-β1 or IL-6 (not shown). Detectable vimentin was present in FaDu cells, although, it was not clearly regulated by TGF-β1 or IL-6 (not shown). In term of IL-6 we recognized an upregulation of vimentin and KLF4 in FaDu-cells, however it was not statistically significant. Slug was less but stable detected in FaDu cells, and it was not statistically significantly increased after TGF-β1 or IL-6 treatment (not shown).

In UPCI-SCC090 cells TGF-β1 induced a moderate, but significant increase (*p* = 0.0499, by Mann Whitney test compared to control; Figure 8a,c) in Slug protein levels, demonstrated in four repeated treatment experiments followed by western blot analysis. The upregulation of Slug by IL-6 was not statistically significant. KLF4 was not significantly regulated by TGF-β1 but was significantly upregulated by IL-6 (*p* = 0.021 by Student’s *t*-test compared to control; Figure 8b,d). A visible, but not significant reduction by TGF-β1 was present (Figure 8b). Vimentin tendentially showed higher protein level by TGF-β1 treatment, but it did not induce significant changes. In contrast, IL-6 treatment induced a significant upregulation of vimentin in UPCI-SCC90 cells (*p* = 0.012 by Student’s *t*-test compared to control; Figure 8e,f). 

Further western blot analysis revealed that TGF-β1 induced phosphorylation and activation of p38 mitogen-activated protein kinase in SCC-25 cells, where the most enhanced TGF-β1 effects were detected (Figure 9c,d). Several treatment repeats demonstrated that activation of the canonical TGF-β1/Smad2/3 pathway was not detectable and thereby activated in SCC-25 and UPCI-SCC-90 cells (not shown). Actually, western-blot bands of Smad2/3 were present, but the phosphorylated Smad 2/3 bands not in control and treated conditions. 

Phospho-Stat3 signaling was detected using chemilumescence by 79.86 kDa, which was activated through IL-6 treatment (Figure 9a,b). 

### 3.5. The Presentation of EMT-Factors KLF4 and Slug Is Focally Modified by Stroma in HNSCC

As previously presented experimentally, stroma-derived cytokines and growth factors as TGF-β1 or IL-6 might modify the protein levels of KLF4 as reduction, and the one of Slug as increase. A comparable situation is detected in HNSCC tissue (Figure 10). Both KLF4 and Slug showed a gradient, a decreasing one of KLF4 from the middle to the border of the tumor cell nest (Figure 10b), and an increasing one of the nuclear reactions of Slug from the middle of the tumor cell nest to the border (Figure 10a). 

## 4. Discussion

The first important finding of this paper is that we confirmed the result of other researchers that KLF4 gene expression was significantly decreased in human cancer tissues compared with normal mucosa controls [11]. This change might be due to hypermethylation of the KLF4 promoter, which contributes to the suppression of KLF4 gene expression [11]. According to Li et al. (2015) [11], the reduction of KLF4 expression is more enhanced in poorly differentiated tumor. 

In parallel to reduction of KLF4 gene expression in HNSCC, the Slug gene expression significantly increased (Figure 1). Moreover, in RNA samples isolated from 80% of the available HNSCC tissues a statistically significant negative correlation was found between KLF4 and Slug gene expression (Figure 2). This negative correlation meant that KLF4 gene expression decreased below, and at the same time, Slug increased above the level of normal mucosa. In this form, KLF4 was replaced by Slug, which we recognize as KLF4/Slug switch. Interestingly, the cases that contained only 1–2% of the KLF4 gene expression levels of the normal mucosa revealed 4–17-fold upregulation of Slug gene expression and frequently contained TP53 mutations including gain-of function mutations (I195T, R248Q). Moreover, the p53 immunhistochemistry pattern was irregular in 20/26 cases with reduced KLF4 gene expression. As shown on Figure 3, the HNSCC cases with irregular p53 immunohistochemistry were related with increased Slug gene expression. In fact, p53 is known as a tumor suppressor, whose original role is to prevent cancer progression by inhibiting proliferation and inducing apoptosis of tumor cells. Several years ago, in 2009, Wang et al. published that wild-type p53 suppresses cancer invasion by inducing Slug degradation, whereas mutant p53 may stabilize Slug protein [38]. This background might also explain the lower Slug gene expression in HPV-positive HNSCC cases. The HPV-positive HNSCC in most of the cases show regular p53 immunohistochemistry, and wild-type p53 sequence in the protein-coding region [20]. 

Experimental results using SCC-25 cells revealed, that a possible pathway of coordinated reduction of KLF4 protein levels, and induction of Slug protein is the effect of the EMT inducer TGF-β1 (Figure 7a–d). This effect was also reproduced in the HPV-positive UPCI-SCC090 cell line (Figure 8a–d). In both, SCC25 and UPCI-SCC090 cells the KLF4/Slug protein changes were accompanied by induction of vimentin protein, which is a classical EMT-event. Interestingly, Katafiasz et al. in 2011 reported that vimentin is upregulated at mRNA level by exogenously overexpressed Slug [15].

In a further step, we investigated the signaling properties of the TGF-β1-induced KLF4-Slug switch. Our results revealed, that TGF-β1 achieved this transcription factor switch from epithelial KLF4 to mesenchymal Slug by activation of the phosphorylation, and increase of the protein levels of p38 mitogen-activated protein kinase (MAPK) (Figure 9). This pathway is an alternative to the canonical TGF-β1-Smad signaling pathway [39] and known as non-canonical TGF-β1- signaling. At the same time, we also observed that IL-6, which partly might be involved in the upregulation of vimentin and Slug, operates through the Stat3-phosphorylation pathway. The TGF-β1-p38 MAPK signaling seems to be engaged with the replacement of KLF4 with Slug, whereas, the IL-6-STAT3 pathway might at a lower level upregulate Slug, but also increase or stabilize KLF4.

Our preliminary unpublished data indicate, that the detection of EMT by simultaneous staining of both cytokeratin and vimentin in significant per cent of cells in cancer cell nests in HNSCC tissue, as well as increased vimentin detection in cancer cell nests do not have any correlation with patient survival. In contrast, Slug positivity in clusters of cells in cancer cell nests or a diffuse Slug staining in cancer cell nests as presented on Figure 6a, is related with lower patient survival and with radiochemotherapy resistance (in press). As previously published by Brabletz et al., Slug is not only a transcription factor induced in relation of EMT, but it has pleiotropic effects, and its significance is beyond the involvement in partial replacement of epithelial phenotype with a mesenchymal one [3]. Moreover, Slug functions as a master regulator and allows cells to enter into the tumor-initiating cell state [7,8]. Our actual results reveal, that the effects of several possible signaling pathways, as the TGF-β1-p38 MAPK, and the IL-6-STAT3 pathway, might accumulate to increase the Slug protein levels in HNSCC tumor cells. The TGF-β1/p38-MAPK/EMT effect was exemplified in SCC25 cells, where Slug was induced by TGF-β1 together with vimentin, and epithelial marker genes together with KLF4 were down regulated or degraded at the protein level (not shown). In this regard, TGF-β1 appears to be the main inducer of the KLF4/Slug switch that contributes to the replacement of the epithelial phenotype with the mesenchymal one. The newly induced Slug contributes directly to the induction of vimentin [15]. Moreover, in the previous paper of Katafiasz et al., the direct downregulation of epithelial proteins, as for example gap junction protein β6 and keratin 4 (which were the most decreased ones), by the overexpressed Slug was reported in UM-SCC-38 oral SCC cell line [15]. Our current results reveal that this not only happens in experimental conditions, but also in patient-derived HNSCC tumor tissue, localized at the border area of the cancer cell nest, in the tumor-stroma interface. In this heterogeneous area the TGF-β1-responding cells are located in the outer ring of the cancer cell nest, and are labelled with Slug antibody, whereas, non-responding cells in the central part of the cancer cell nest are Slug-negative and contain KLF4 (Figure 10). 

## 5. Conclusions

In RNA samples isolated from 80% of the available HNSCC tissue samples a statistically significant negative correlation was found between KLF4, and Slug gene expression. This negative correlation meant, that KLF4 gene expression decreased under the level of normal mucosa, and at the same time, Slug increased above the level of normal mucosa. In this form, KLF4 was replaced by Slug. In contrast, in 20% of the investigated HNSCC samples this negative correlation was not present. Slug gene expression was present in normal mucosa and in HNSCC, nevertheless, in HNSCC Slug was upregulated. The upregulation was related with induction of EMT by TGF-β1 or IL-6 or with sequence mutation of the p53 coding region or with loss of the p53 gene product.

## Figures and Tables

**Figure 1 cells-10-00539-f001:**
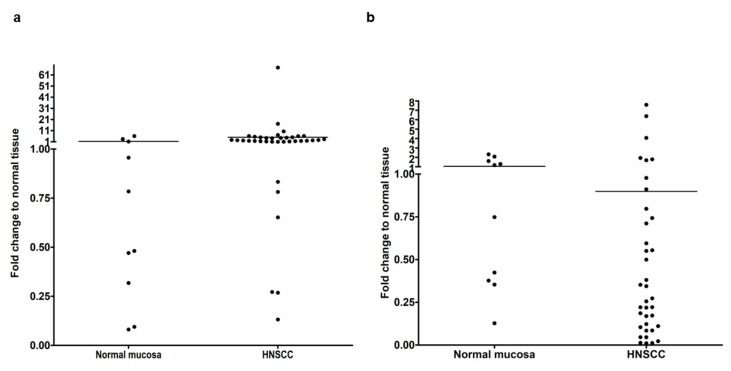
Comparison of relative quantification of Slug (**a**) and KLF4 (**b**) gene expression in normal mucosa and in HNSCC. Ten normal mucosa and 37 HNSCC samples were used for real-time PCR analysis. On the *Y*-axis the fold change difference to the mean reference value of normal mucosa is presented. None of the compared sample sets showed normal distribution. The medians of control and HNSCC samples were compared using Mann-Whitney-test. Slug gene expression was significantly higher in HNSCC than in normal mucosa (*p* = 0.009), whereas, KLF4 gene expression was significantly lower in HNSCC than in normal mucosa (*p* = 0.041).

**Figure 2 cells-10-00539-f002:**
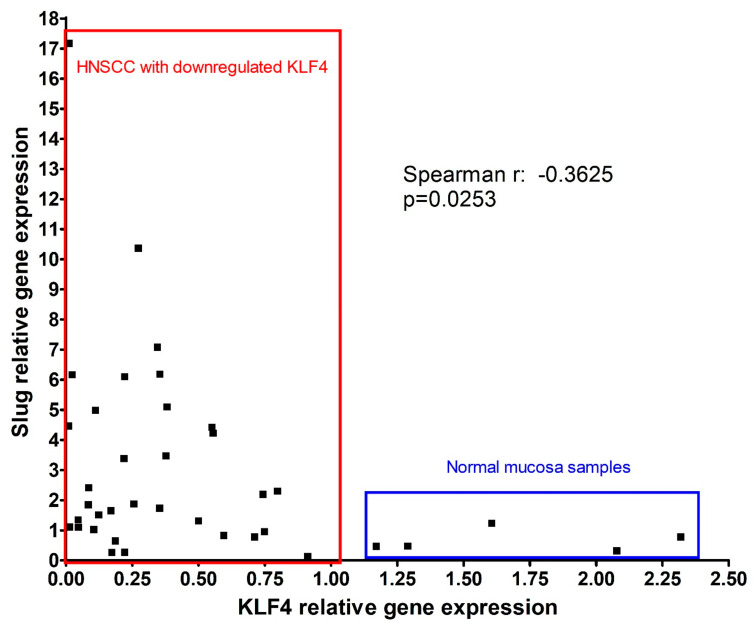
In HNSCC where KLF4 is reduced (red box) compared to normal mucosa from UPPP (blue box), Slug gene expression is upregulated. HNSCC with reduced KLF4 gene expression have a negative correlation between KLF4 and Slug gene expression.

**Figure 3 cells-10-00539-f003:**
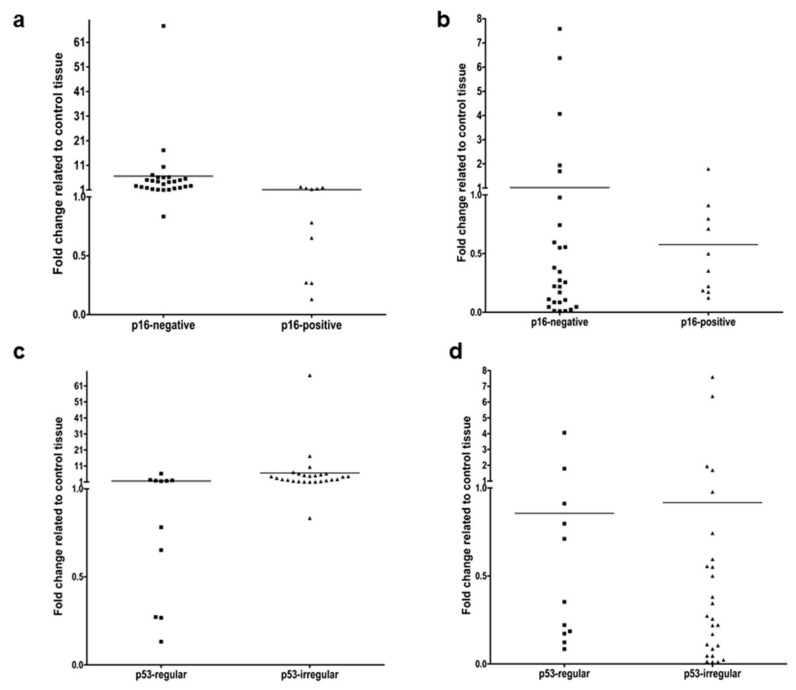
Comparison of relative quantification of Slug (**a**,**c**) and KLF4 (**b**,**d**) gene expression in p16-positive and negative (**a**,**b**) HNSCC, as well as in HNSCC with regular and irregular p53 gene expression (**c**,**d**). Ten HPV-positive and 27 HPV-negative HNSCC samples were used for real-time PCR analysis. On the *Y*-axis the fold change difference to the mean reference value of normal mucosa is presented. None of the compared sample sets showed normal distribution. The median of Slug gene expression in p16-negative HNSCC samples was significantly higher (*p* = 5 × 10^−4^) than in p16-positive samples (**a**). In contrast, KLF4 gene expression was not significantly different in p16-positive or negative cases (**b**). Eleven HNSCC samples were available with regular (wild type) and 26 with irregular (mainly mutated) p53 gene background. The median of Slug gene expression in HNSCC samples with irregular p53 was significantly higher (*p* = 5.5 × 10^−3^) than in samples with regular p53 (**c**). In contrast, KLF4 gene expression was not significantly different in cases with regular or irregular p53 (**d**). All datasets were not normal distributed, and Mann-Whitney-test median comparison was used.

**Figure 4 cells-10-00539-f004:**
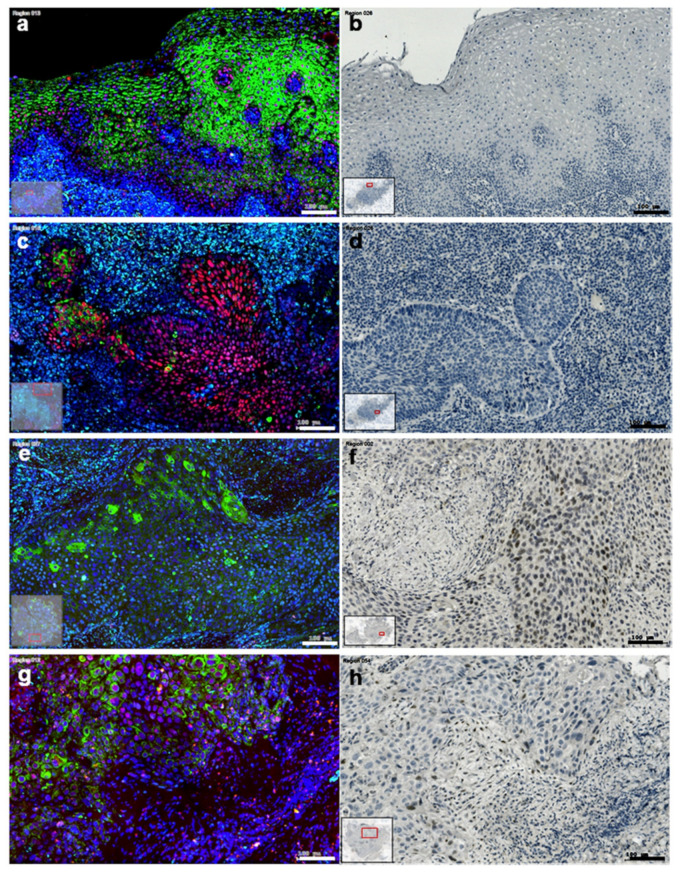
Comparison of immunohistochemical labeling of KLF4 (red), pan-cytokeratin (green) and vimentin (light blue) (**a**,**c**,**e**,**g**) with enzyme immunohistochemistry of Slug (**b**,**d**,**f**,**h**) in normal layered epithelium (**a**,**b**); in HPV^+^, p53 wild type HNSCC (**c**,**d**); in HNSCC with loss of p53 gene product (**e**,**f**) and in HNSCC with p53 gain of function mutation (**g**,**h**). The normal layered epithelium showed dispersed intensive green cytokeratin (CK) reaction, the nuclei of the CK^+^ cells contained KLF4, the CK^−^ stroma region contained vimentin-positive (VIM^+^) cells (**a**). In this tissue no traces of Slug staining were detected (**b**). The HPV^+^ p53 wild type HNSCC showed limited green cytokeratin (CK) reaction, the nuclei in cancer cell nests contained KLF4, also if they were CK^−^, the stroma region contained numerous vimentin-positive (VIM^+^) cells (**c**). In this tissue no traces of Slug staining were detected (**d**). In the HNSCC with loss of p53 gene product diffuse CK reaction was detected at different levels in cells of the cancer cell nests, only few scattered nuclei in cancer cell nests contained red KLF4 reaction, the light blue reaction of VIM was spread in the stroma, but traces of VIM were also visible in the cancer cell nests (**e**). In this tissue intensive diffuse Slug reaction was detected (**f**). In HNSCC with p53 gain of function mutation diffuse CK reaction was detected at different levels in cells of the cancer cell nests, the majority of the nuclei in cancer cell nests contained red KLF4 reaction, and in this tissue section the vimentin reaction was scattered (**g**). Several cell nuclei at border areas of cancer cell nests were Slug^+^ (**h**).

**Figure 5 cells-10-00539-f005:**
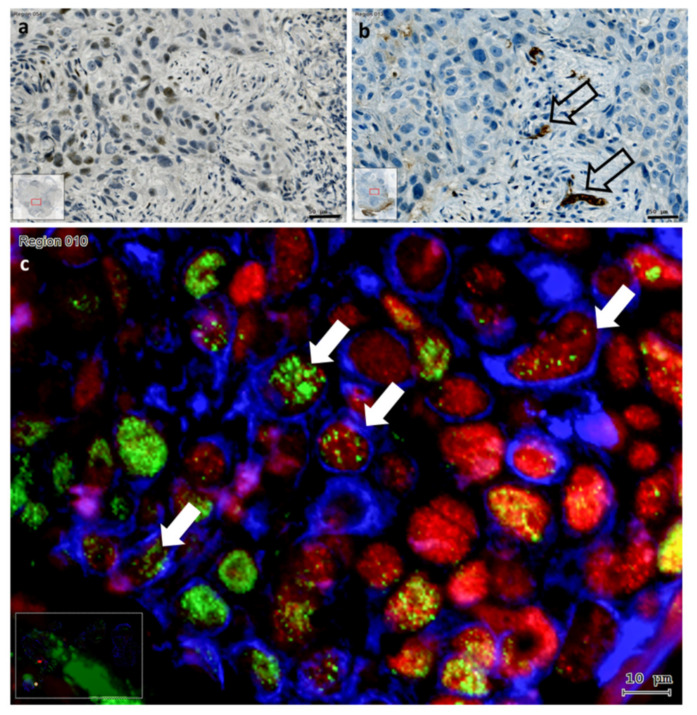
Comparison of enzyme immunohistochemical labeling of Slug (**a**) and TGF-β1 (**b**) with immunofluorescence labeling of KLF4 (red), pan-cytokeratin (blue) and vimentin (green) combined (**c**) in HNSCC with p53 gain of function mutation. The Slug^+^ cells formed a cluster in the cancer cell nest in the tumor-stroma interface (**a**). In this location scattered stroma cells, and at a lower extent also tumor cells reacted positively with antibody against TGF-β1 (open big arrows) (**b**). Using increased magnification acquisition of the combined immunofluorescence labeling of CK, vimentin and KLF4 (**c**), more cells with combined CK, KLF4 and vimentin were detected (**c**) (white arrows). Bars: 50 µm in brightfield and 10 µm in fluorescence. The colors in fluorescence were chosen to present the co-localizations optimally.

**Figure 6 cells-10-00539-f006:**
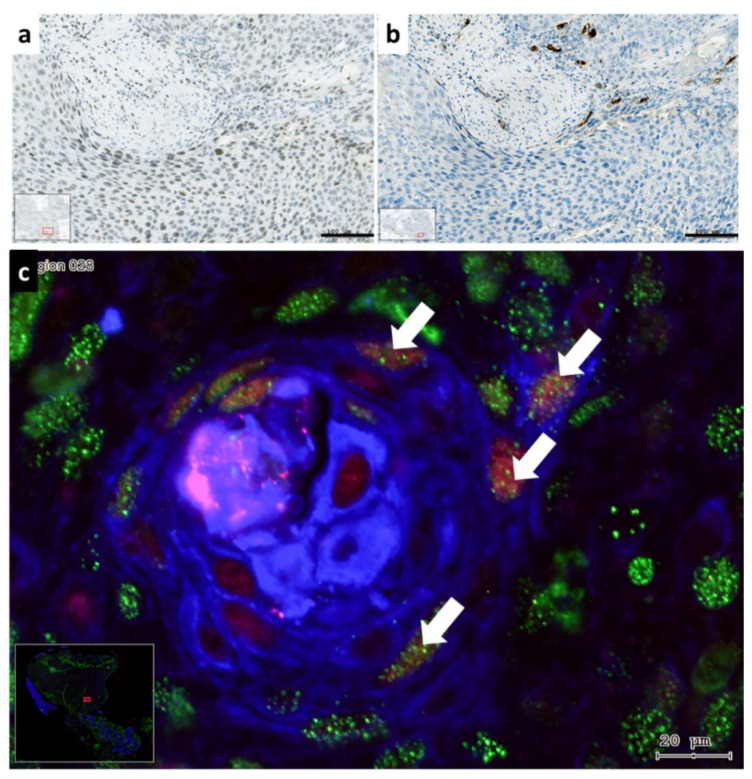
Comparison of enzyme immunohistochemical labeling of Slug (**a**) and TGF-β1 (**b**) with immunofluorescence labeling of KLF4 (red), pan-cytokeratin (blue) and vimentin (green); combined (**c**) in HNSCC with loss of p53 gene product. The Slug^+^ cells were diffusely present in the cancer cell nest (**a**). In this location scattered stroma cells reacted positively with antibody against TGF-β1 (**b**). Using increased magnification acquisition of the combined immunofluorescence labeling of CK, vimentin and KLF4 (**c**), more cells with combined KLF4, vimentin and CK (**c**), were detected (white arrows). Bars: 100 µm in brightfield and 20 µm in fluorescence. The colors in fluorescence were chosen to present the co-localizations optimally.

**Figure 7 cells-10-00539-f007:**
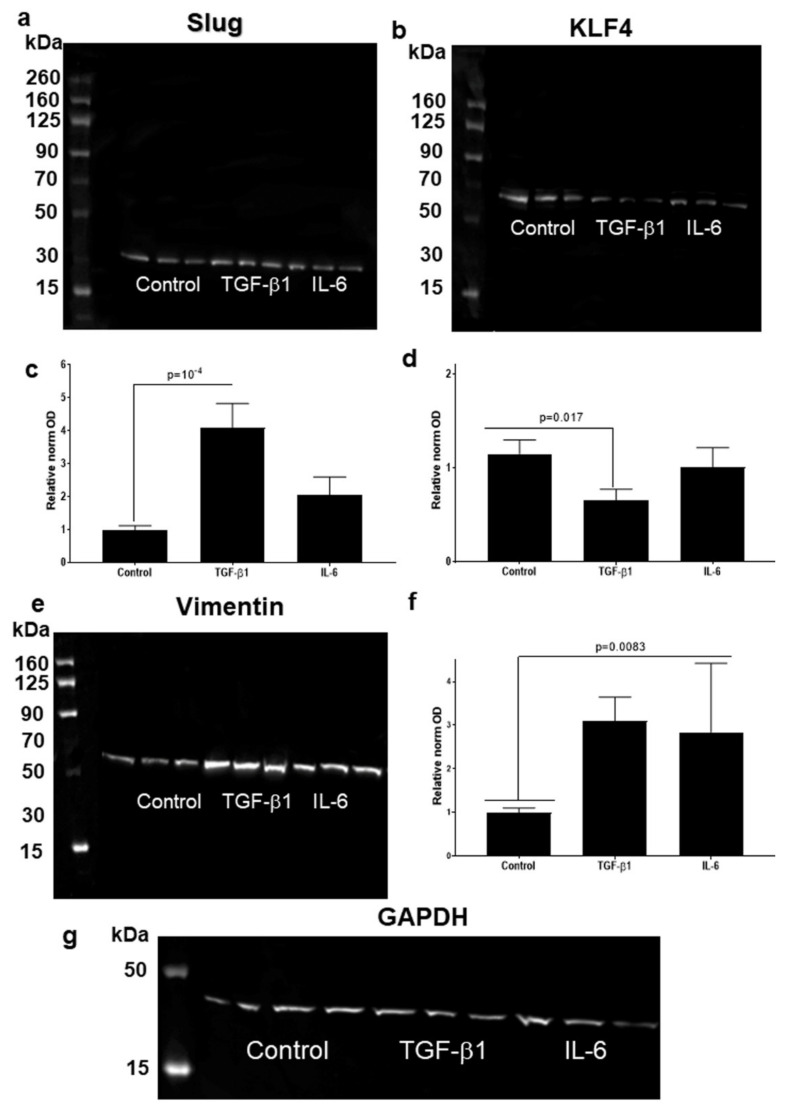
Western blot detection of Slug (**a**), KLF4 (**b**) and vimentin (**e**) proteins in SCC-25 cells in control, 1 ng/mL TGF-β1 and in 50 ng/mL IL-6 treated conditions. Both Slug (**c**) (*p* = 10^−4^ by Kruskal-Wallis test and Dunn’s multiple comparison) and vimentin (**f**) (*p* = 0.0083 Kruskal-Wallis test and Dunn’s multiple comparison) showed significant upregulation by TGF-β1 treatment (**a**,**c**,**e**,**f**), whereas the KLF4 protein levels (**b**,**d**) were statistically significantly reduced (*p* = 0.017 by unpaired *t*-test compared to control). IL-6 upregulated Slug (**c**) and downregulated KLF4 (**b**,**d**), but it was not statistically significant. Vimentin showed a significant increase (*p* = 0.0083 Kruskal-Wallis test and Dunn’s multiple comparison) by IL-6 treatment (**e**,**f**). Four western blot membranes were acquired digitally, and the band optical densities (ODs) of proteins of interest and of loading control (GAPDH) were measured using the Image Studio Lite of Li-cor. The ODs of proteins of interest were normalized to loading control. Mean normalized optical densities of control samples were set to “1”, and this control mean was used as reference (**c**,**d**,**f**). Loading control by western blot membrane reacted with GAPDH antibody is presented on panel (**g**).

**Figure 8 cells-10-00539-f008:**
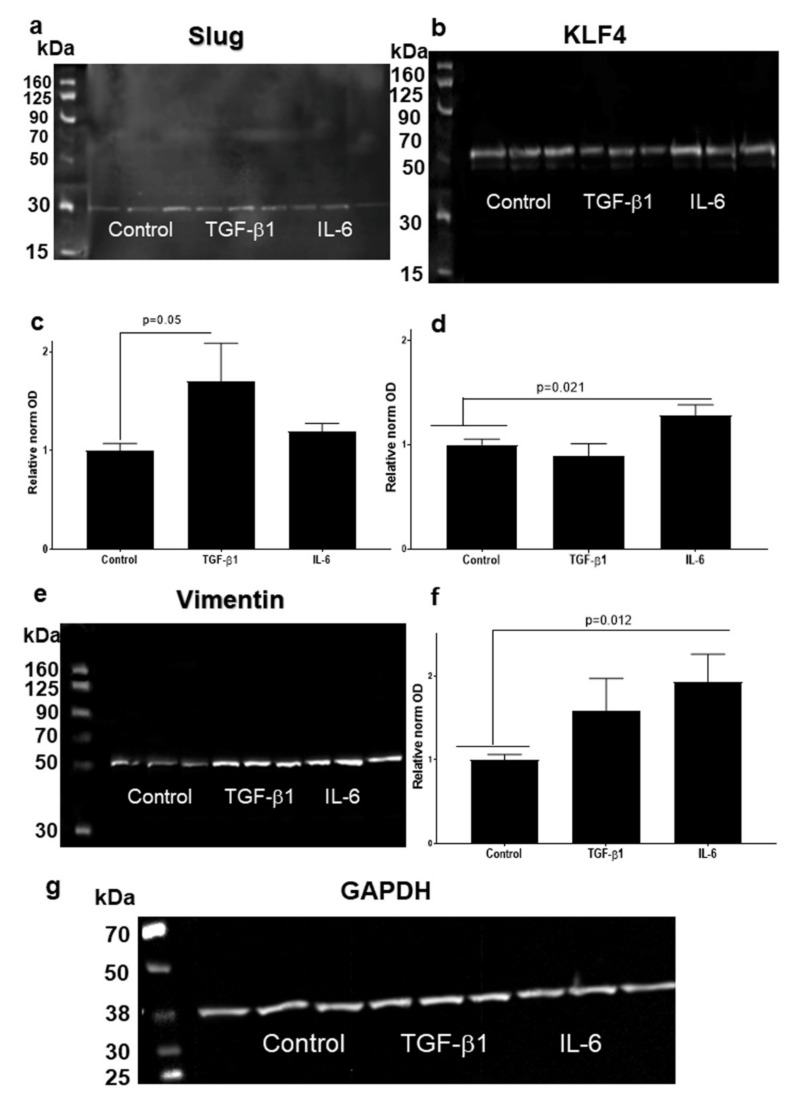
Representative western blots of Slug (**a**), KLF4 (**b**) and vimentin (**e**) proteins in UPCI-SCC-90 cells in control, 1 ng/mL TGF-β1 and in 50 ng/mL IL-6 treated conditions. TGF-β1 induced a moderate but significant increase (*p* = 0.0499, by Mann Whitney test compared to control) in Slug protein level. Il-6 did not statistically upregulated Slug in UPCI- SCC-90 cells (**a**,**c**). KLF4 was not regulated by TGFβ1, but it was upregulated by IL-6 (*p* = 0.021 by Student’s *t*-test compared to control) (**b**,**d**). TGF-β1 upregulated vimentin at protein levels, but it was not statistically significant. IL-6 induced significant upregulation of vimentin (*p* = 0.012 by Student’s *t*-test compared to control) (**e**,**f**). Four western blot membranes were acquired digitally, and the band optical densities (ODs) of proteins of interest and of loading control (GAPDH) were measured using the Image Studio Lite of Li-cor. The ODs of proteins of interest were normalized to loading control. Mean normalized optical densities of control samples were set to “1”, and this control mean was used as reference (**c**,**d**,**f**). Loading control by western blot membrane reacted with GAPDH antibody is presented on panel (**g**).

**Figure 9 cells-10-00539-f009:**
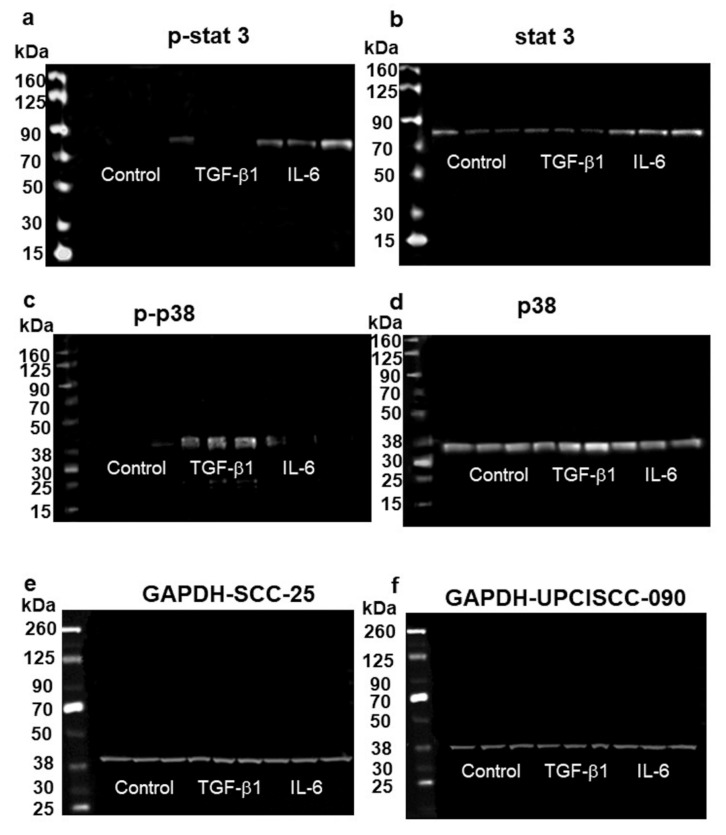
Western blot detection of p-stat3 (**a**), stat3 (**b**), p-p38 (**c**), p38 (**d**) proteins in SCC-25 cells in control, 1 ng/mL TGF-β1 and in 50 ng/mL IL-6 treated conditions. GAPDH was used as loading control in both SCC-25 and UPCI-SCC90 cells (**e**,**f**). Phospho-stat3 (p-stat3) and stat-3 (**a**,**b**) were upregulated by IL-6 treatment. TGF-β1 induced phosphorylation and activation of the estimated p38 mitogen-activated protein kinase (**c**,**d**).

**Figure 10 cells-10-00539-f010:**
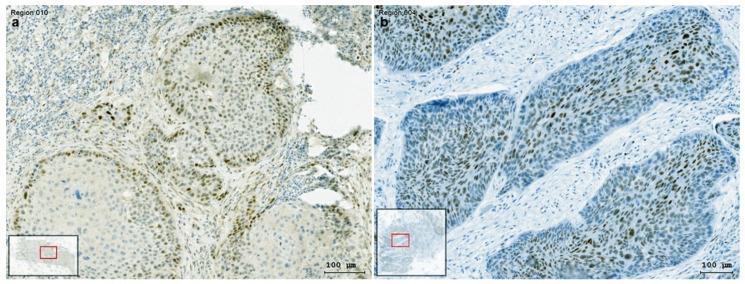
Enzyme immunohistochemistry for Slug (**a**), and KLF4 (**b**) in HNSCC tumor tissue. Staining differences between the core and the border of the tumor cell nest were analyzed by TissueFax^®^ (Tissuegnostics^TM^). Slug IHC reaction was detected at the border of the tumor cell nest (**a**). The tumor cell nest boundary showed a drastic decline or loss of the epithelial marker KLF4 in Slug positive HNSCC tumor tissue samples (**b**). The decreasing gradient of KLF4 from the middle to the border of the tumor cell nest (**b**), was presented together with the increasing gradient of the nuclear reaction of Slug, localized at the border of the tumor cell nest (**a**).

**Table 1 cells-10-00539-t001:** Study population: Characteristics of the 37 patients with head and neck cancer, who gave their written informed consent to spend a histological sample for this study, and the RNA and cDNA samples were at sufficient quality for further processing.

**sex**	male	36	97%
	female	1	3%
**age**	≤50	1	3%
	51–60	7	19%
	61–70	21	57%
	71–80	4	11%
	>80	4	11%
**ASA score**	ASA I/II	26	70%
	ASA III/IV	11	30%
**tumor site**	lips/oral cavity	4	11%
	oropharynx	16	43%
	hypopharynx	7	19%
	Larynx	10	27%
**histology**	squamous cell carcinoma	37	100%
**HPV/tumor localization**	P16-positive oropharynx SCC	9	56.25%
	P16-negative oropharynx SCC	7	43.75%
	P16-positive larynx SCC	1	10%
	P16-negative larynx SCC	9	90%
	P16-negative SCC of lips/oral cavity	4	100%
	P16-negative SCC of hypopharynx	7	100%
**UICC stage**	stage 1	4	11%
	stage 2	2	5%
	stage 3	11	30%
	stage 4a	17	46%
	stage 4b	2	5%
	stage 4c	1	3%
**HPV status**	negative (<70%)	27	73%
	positive (≥70%)	10	27%
**P53 status**	regular	11	30%
	irregular	26	70%
**treatment modality**	surgery	9	24%
	surgery & RT	7	19%
	surgery & RCHT/RIT	1	3%
	RCHT/RIT	18	49%
	RT	1	3%
	CHT	1	3%

**Table 2 cells-10-00539-t002:** Overview of primary antibodies for immunohistochemistry and immunofluorescence staining.

Antibody	Isotype	Catalogue Number	Manufacturer	Dilution
pan-cytokeratin	IgG1	760-2595	Roche Ventana, Mannheim, Germany	pre-diluted
anti-vimentin, (clone VI-10)	IgM	EX-11-460-C100	Exbio, Prague, Czech Republic	1:200
anti-KLF4	IgG	ab215036	Abcam, Cambridge, UK	1:2000
anti-slug	IgG1	564614	BD Pharmingen Austria, Vienna Austria	1:400
anti-TGFβ1	IgG	AB190503	Abcam, Cambridge, UK	1:100
Isotype-control mouse	IgG1	11-632-C100	Exbio, Prague, Czech Republic	1:100
Isotype-control mouse	IgM	11-803-C100	Exbio, Prague, Czech Republic	1:100
Isotype-control rabbit	IgG	02-6102	Invitrogen Life Technologies, Darmstadt, Germany	1:400

**Table 3 cells-10-00539-t003:** Overview of primary utilized antibodies for western blot.

Antibody	Isotyp	Catalogue Number	kDa	Manufacturer	Dilution
anti-Slug	IgG1	9585	30	Cell Signalling Technologies, Danvers, MA, USA	1:1000
anti-Slug	IgG1,_K_	564614	30	BD Pharmingen, Szabo Scandic, Vienna Austria	1:400
anti-vimentin (clone SP-20)	IgG	M3202	53	Spring Biotech, Linaris, Germany	1:100
anti-GAPDH	IgG1	ab8245	36	Abcam, Cambridge, UK	1:5000
anti-KLF4	IgG1	ab215036	54	Abcam, Cambridge, UK	1:1000
anti-p38 MAPK	IgG	8690S	40	Cell Signalling Technologies, Danvers, MA, USA	1:1000
anti-phospho-p38 MAPK	IgG	4511S	43	Cell Signalling Technologies, Danvers, MA, USA	1:1000
anti-Smad2/3	IgG	3102	52.6	Cell Signalling Technologies, Danvers, MA, USA	1:1000
anti-phospho-Smad2	IgG	3108	60	Cell Signalling Technologies, Danvers, MA, USA	1:1000
anti-Stat3	IgG	12640	79.86	Cell Signalling Technologies, Danvers, MA, USA	1:1000
anti-phospho-Stat3	IgG	9145	79.86	Cell Signalling Technologies, Danvers, MA, USA	1:2000

## Data Availability

All data supporting reported results is contained within the article or Appendix A.

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
