# Peer review of "KLF4, Slug and EMT in Head and Neck Squamous Cell Carcinoma"

_cells, 2021, doi:10.3390/cells10030539_

Round 1
Reviewer 1 Report
I consider that the authors have addressed all my concerns and that the manuscript can be published.
Author Response
Reviewer-1 has no concerns, suggests publishing.
Answer:
Authors are grateful for the efforts and the work of Reviewer 1.
Reviewer 2 Report
The manuscript describes an interesting topic of epithelial-mesenchymal transition (EMT) in HNSCC. The EMT process plays an important role in initiation, primary tumor growth, invasion, dissemination and metastasis, as well as acquisition of therapeutic resistance. The authors clarify the expression and role of important EMT factors - KLF4, SNAI2 at the protein and mRNA level.
The work has several shortcomings and requires a major revision.
Some parts of the article are described in too much detail (Material and Methods section), with a non-standard description (90% DMEM/F12 medium, 90% EMEM).
On the contrary, other parts are not sufficiently elucidated. The section Introduction lacks a description of the role of HPV infection in HNSCC, although the detection of HPV positivity / negativity by p16 is described in the section Results. Information on p53 protein and role of TGF beta effects in HNSCC is also lacking.
The M&M section incorrectly describes the immunofluorescence method in the subtitle Immunohistochemistry. A description of the western blot analysis is included in the Cell treatments section.
For better clarity and orientation, it would be appropriate to rank the antibodies and their dilution used in Immunofluorescence and Immunohistochemistry as well as in Western blot in table.
In the Results section:
The issue is the use and presentation of results using non-specific primers in the results section. Verification of primers for qPCR analysis is a basal step before experimental work. The KLF4 gene expression analysis was done from 39 samples of HNSCC, expression of SNAI2 was analyzed in 37 samples. Specify, please, the total number of analyzed samples. The SNAI2 and Slug designations need to be unified and a single designation needs to be used.
The names of the individual subheadings in the section Results are misleading and do not reflect the results described (3.3. Comparison of Slug and KLF4 protein with TGF-b in HNSCC; 3.4. Interaction of Slug, vimentin and KLF4 with TGF-b in experimental HNSCC cell line models).
Western blot analyzes lack an image of GAPDH levels as a loading control.
Images of immunofluorescence / immunohistochemical labeling are not sufficiently described, especially in the sections describing multiple labeling of proteins. Figure 6c does not clearly demonstrate the combined labeling of KLF4 and vimentin. Other confirmatory methods should be used to demonstrate colocalization of proteins.
It is necessary to improve the characteristics of patient HNSCC samples - HPV status for individual types of tumors. I consider the use of oropharyngeal mucosa as a control for all types of HNSCC to be a significant lack of work.
The article needs to be revised and the conclusions resulting from the individual experiments described more clearly.
Author Response
Reviewer comments and answers:
Some parts of the article are described in too much detail (Material and Methods section), with a non-standard description (90% DMEM/F12 medium, 90% EMEM).
Answer:
Authors are grateful for the constructive comments of the Reviewer, and for the valuable suggestions. The above suggested reduction of excess details was completed and the text became focused. (Page 6, lines 208, 210; pages 4-5, part 2.2 and 2.3)
On the contrary, other parts are not sufficiently elucidated. The section Introduction lacks a description of the role of HPV infection in HNSCC, although the detection of HPV positivity / negativity by p16 is described in the section Results. Information on p53 protein and role of TGF beta effects in HNSCC is also lacking.
Answer:
Authors completely agree with this point, the required text is now included in the Introduction. (Page 2, lines 66-86)
The M&M section incorrectly describes the immunofluorescence method in the subtitle Immunohistochemistry. A description of the western blot analysis is included in the Cell treatments section.
Answer:
Authors completely agree with this point, the required correction is now included in the M&M section. (Page 5, line 158 – Page 6 line 201; Page 7, line 233 – Page 8, line 274).
For better clarity and orientation, it would be appropriate to rank the antibodies and their dilution used in Immunofluorescence and Immunohistochemistry as well as in Western blot in table.
Answer:
Authors completely agree with this point, the required tables have been now included as Tables 2 and 3. (Pages 6, and 8).
In the Results section:
The issue is the use and presentation of results using non-specific primers in the results section. Verification of primers for qPCR analysis is a basal step before experimental work. The KLF4 gene expression analysis was done from 39 samples of HNSCC, expression of SNAI2 was analyzed in 37 samples. Specify, please, the total number of analyzed samples. The SNAI2 and Slug designations need to be unified and a single designation needs to be used.
Answer:
All these suggestions are relevant, were completed and fulfilled as suggested. Both, the gene expression analysis of Slug and KLF4 was done from overall 37 samples of HNSCC. (Page 8, line 277). Now, only the Slug designation is used in the paper. All technical supporting data are removed from the main text and provides as “Technical supporting data”.
The names of the individual subheadings in the section Results are misleading and do not reflect the results described (3.3. Comparison of Slug and KLF4 protein with TGF-b in HNSCC; 3.4. Interaction of Slug, vimentin and KLF4 with TGF-b in experimental HNSCC cell line models).
Answer:
Accordingly, subheadings have been revised. (Page 12, line 377, Page 16, line 459-461 and 408/409). Similarly, all other subheadings are revised and corrected.
Western blot analyzes lack an image of GAPDH levels as a loading control.
Answer:
The GAPDH images are now included in Figures 7-8. (Pages 17 and 18).
Images of immunofluorescence / immunohistochemical labeling are not sufficiently described, especially in the sections describing multiple labeling of proteins. Figure 6c does not clearly demonstrate the combined labeling of KLF4 and vimentin. Other confirmatory methods should be used to demonstrate colocalization of proteins.
Answer:
Multi-labelled immunofluorescence images have been replaced by better ones. (Pages 14 and 15, Figure 5c and Figure 6c). The vimentin – cytokeratin co-localization has been also confirmed by confocal microscopy and published by us previously elsewhere (Steinbichler et al. J. Clin. Med. 2020, 9, 2061; doi:10.3390/jcm9072061.)
It is necessary to improve the characteristics of patient HNSCC samples - HPV status for individual types of tumors. I consider the use of oropharyngeal mucosa as a control for all types of HNSCC to be a significant lack of work.
Answer:
The “Table 1: Study population” was revised accordingly, as suggested. (Pages 3 and 4)
The article needs to be revised and the conclusions resulting from the individual experiments described more clearly.
Answer:
As suggested, the whole article has been extensively revised. We would like to thank for suggestions and ideas for improvement.
Reviewer 3 Report
The authors measured the mRNA levels of EMT-related TFs (SNAI1, 2, ZEB1, TWIST and KLF4) in 47 HNSCC and 10 control mucosal tissues and found that SNAI2 (encoding Slug) was upregulated and KLF4 was downregulated in HNSCC significantly. There was a negative correlation between SNAI2 and KLF4 mRNA levels in KLF4-downregulated HNSCC samples. SNAI2 upregulation was associated with p16 negativity (associated with HPV negativity) and altered p53 staining pattern (associated with TP53 mutation) whereas KLF4 downregulation did not show such association. They also visualized the expression of SNAI2, KLF4, TGF-beta, vimentin with IF and/or IHC in HNSCC tissues. In vitro, they found that TGF-beta and IL-6 variably induced SNAI2 and downregulated KLF4 expression in cell lines. In general, the study did not provide any novel finding or mechanistic insight. The reviewer therefore recommends rejection of their manuscript.
Concerns:
- The clinical significance of SNAI2 upregulation and KLF4 expression in HNSCC has been well reported (Cell Adh Migr 2011;5(4):315-22; Oral Oncol 2020;111:104948.; Cancer Sci. 2011;102(4):895-902.). In the present study, the author only measured the mRNA expression level of SNAI2 and KLF4 without analysing their correlation with cancer recurrence or patients’ survival.
- The authors used p16 and p53 staining as surrogate markers of HPV status and p53 mutation status, respectively. They then inferred the correlation of SNAI2 upregulation/KLF4 downregulation with HPV status and p53 mutation status using these markers. In essence, they are reporting correlations with correlated variables instead of the true HPV and p53 mutation status, which is not scientifically sound.
- The authors did not perform any quantitative analysis of IHC/IF.
- Regulation of SNAI2 by TGF-beta is well known.
- The data is also poorly presented throughout the manuscript.
Author Response
Reviewer comments and answers:
- The clinical significance of SNAI2 upregulation and KLF4 expression in HNSCC has been well reported (Cell Adh Migr 2011;5(4):315-22; Oral Oncol 2020;111:104948.; Cancer Sci. 2011;102(4):895-902.). In the present study, the author only measured the mRNA expression level of SNAI2 and KLF4 without analysing their correlation with cancer recurrence or patients’ survival.
Answer:
Authors are grateful for the suggestions for additional high relevant literature sources, which are now included in the Introduction and the Discussion. (Page 2, lines 53-54; lines 72-76; page 21, lines 580-581, 605-610). The following suggested reference we also included in the previous submission: Cancer Sci. 2011;102(4):895-902. These works are valuable, interesting, and significantly contributed to the field, moreover, they were also instrumental for the improvement of the Discussion section of the article. In our study we defined a clear hypothesis in the Introduction and it does not overlap with any of these previous publications. In contrast to the study mentioned by the Reviewer: Tai et al. Cancer Sci. 2011;102(4):895-902, in our results, KLF4 representation in the tumor tissue and it´s gene expression did not show a significant relationship with patients´survival.
In contrast, Slug representation in % of cells in epithelial cancer cell nests had major clinical relevance, which is the focus of a separately submitted manuscript.
The main finding was: when treated with primary RT/CRT Slug-positive patients responded worse and had a worse 5-year overall survival (OS) than Slug-negative patients.
A detailed manuscript of the clinical data has been submitted elsewhere.
Authors find that an opposite regulation of Slug and KLF4 by TGF-beta-1 produced in the microenvironment of the cancer cell nests is a novel finding.
Taken together, Authors are grateful for the reviewer for suggesting valuable references for improving the Discussion part.
- The authors used p16 and p53 staining as surrogate markers of HPV status and p53 mutation status, respectively. They then inferred the correlation of SNAI2 upregulation/KLF4 downregulation with HPV status and p53 mutation status using these markers. In essence, they are reporting correlations with correlated variables instead of the true HPV and p53 mutation status, which is not scientifically sound. -
Answer:
The concerns of the Reviewer are recognized by the authors, please find our response as follows:
The sensitivity of p16-IHC is 78% and the specificity is 79% for the determination of HPV status, whereas, in present day PCR is also performed routinely in order to confirm HPV in HNSCC tissue. Nevertheless, the favorable prognosis is based on the p16-IHC-status and not on the HPV-PCR.
See also: “The 5-year OS of the HPV-/p16+ HNSCC was intermediate while HPV+/p16- and HPV-/p16- had the shortest survival outcomes.” Reference: Albers et al. Sci Rep. 2017 Dec 1;7(1):16715. doi: 10.1038/s41598-017-16918-w.
See also: Stephen et al. Cancer Clin Oncol . 2013;2(1):51-61. doi: 10.5539/cco.v2n1p51.
P53 might be lost or stabilized (increased stained) at protein level and mutation status is associated with an extended immunohistochemistry pattern. A strong p53 IHC might mean protein stabilization or mutation. No staining might be due to poor detection of the wild type p53 protein, which is quickly degraded in the cells, but also to protein lost. In this regard, we sequenced 29 HNSCC cases for the mutation status in the protein-coding region. Seventeen of them showed wild type sequence, 11 showed the following p53 mutations: P151H, E285K, Y220C, R273P, I195T, A129G – L130F in the same patient, V197E, R248Q, R273C, R273H, G262V. In one patient p53 mRNA was lost.
In p53-mutated or mRNA-lost cases Slug showed a significant (p=0.0021) upregulation, exactly as presented in the manuscript (Page 11, Figure 3 c).
In case of KLF4, the p53-mutated or mRNA-lost cases did not display significantly different gene expression compared to the p53 wild type cases, exactly as presented in the manuscript (Page 11, Figure 3 d).
- The authors did not perform any quantitative analysis of IHC/IF.
Answer:
Both KLF4 and Slug has been stained with enzymatic immunohistochemistry on a Ventana immunostainer utilizing the DAB-MAP kit. Both the stainings were quantified by % of positive cells. The negative relationship between the % of positive cells showed also a tendentially negative relationship between KLF4 and Slug.
- Regulation of SNAI2 by TGF-beta is well known.
Answer: Due to different backgrounds of the used cell lines including the HPV-positive oropharynx carcinoma cell line UPCI-SCC090, the response to TGF-beta1 was different in the cell lines. Activation of several regulatory pathways by TGF-beta1 were hypothesized by the authors, and both the canonical and the non-canonical pathways, as well as the co-activation of STAT3 (by investigating the phosphorylation of Smads, STAT3 and p38) were checked by repeated western blots, until finally the activation of the non-canonical pathways could be described in the results, and discussed.
- The data is also poorly presented throughout the manuscript.
Answer: Authors agree, that the presentation level required improvement, which is done in the revised version.

Round 2
Reviewer 3 Report
The concerns over the novelty and data presentation issues remain unresolved. I still recommend rejection.
Author Response
In our study we set up a clear hypothesis (introduction part), and it does not overlap with any other already known results, or theses of previous publications.
Several transcription factors (TFs) have the potential to regulate EMT. In this manuscript we investigated the gene expression of Snail, Slug, TWIST, ZEB1 and KLF4 in a representative sample of HNSCC patient tissues, and found that both Slug and KLF4 are represented in significant quantities in total RNA extracts of HNSCC patient tissue samples. Our previous publication (Steinbichler et. al, J Clin Med 2020, 9) described Slug as a surrogate marker of EMT. Here we confirmed this at mRNA level, and found that other TFs as TWIST, Snail, and ZEB1 are less abundant in HNSCC tissue. Based on previous literature background, an opposed relationship between Slug and KLF4 was hypothesized. The negative relationship between the mRNA expression of Slug and KLF4 was evidenced as a significant negative correlation in 80% of the HNSCC cases. Slug was significantly higher expressed in HPV-negative HSNCC samples and in the ones with irregular p53 coding sequence. From our immunohistochemical data it was obvious, that Slug was frequent in tumors with irregular p53 or p53 loss, which revealed that wild type p53 might be an inhibitor of Slug. The degradation of Slug in p53-wild type background has been published before (Wang et al. Nat Cell Biol 2009, 11, 694-704). Our data indicate that the inhibition might also occur at mRNA/transcriptional level.
HNSCC tissue is characterized by heterogeneity in the cancer cell nests. Our results indicate that Slug might be induced in tumor cells at the border of cancer cell nests in response to stroma-derived TGF-beta1. Immunohistochemical data revealed that in Slug-positive HNSCC with irregular p53, EMT occurs in response to TGF-beta1, whereas, high stroma TGF-beta1 production alone is not enough for EMT and Slug induction in HNSCC with wild-type p53.
Experimental results revealed that both in p53-wild type (and HPV+) and in p53-irregular (and HPV-) HNSCC cell lines TGF-beta1 managed to achieve the KLF4/Slug replacement, which might be an important event in EMT of HNSCC. We also investigated the corresponding mechanism, and identified the non-canonical p38-based TGF-beta1 signaling pathway.
A particularly interesting finding in our study was, that the KLF4/Slug switch, which is regulated by the non-canonical p38-based TGF-beta1 signaling pathway, could be also detected in the HNSCC tissue (Figure 10), and allowed a translational connection of the benchmark data to clinical context.
We have just published that Slug overexpression can be used as predictive marker, in particular, patients who overexpress Slug had a 3.3 times better chance of survival when treated with upfront surgery/PORT versus primary RT/CRT (Riechelmann et al. Cancers, 2021, 13, 772). Our results in this actual submission to Cells are our first efforts to get the biomechanistic background for this exceptionally important clinical finding. Two major effects, irregular p53 and TGF-beta1 seem to be critical factors in regulation of KLF4/Slug switch, which is so important that we now rely on Slug overexpression for therapy decisions for HNSCC.
In our opinion: our manuscript contains several novel data and contains clinically relevant experimental findings that are also translated into investigation of patient tissues.